# Social Capital Associated with Quality of Life among People Living with HIV/AIDS in Nanchang, China

**DOI:** 10.3390/ijerph16020276

**Published:** 2019-01-18

**Authors:** Fei Xie, Huilie Zheng, Ling Huang, Zhaokang Yuan, Yuanan Lu

**Affiliations:** 1Jiangxi Province Key Laboratory of Preventive Medicine, School of Public Health, Nanchang University, Nanchang 330006, China; 406530516742@email.ncu.edu.cn (F.X.); 13319408889@163.com (H.Z.); hl408180964@163.com (L.H.); 2Center for Disease Prevention and Control of Pudong New District, Shanghai 200136, China; 3Department of Public Health Sciences, University of Hawaii at Manoa, Honolulu, HI 96822, USA

**Keywords:** HIV/AIDS, social capital, quality of life, Personal Social Capital Scale (PSCS), Nanchang, China

## Abstract

*Background*: This study aims to explore the relationship between quality of life (QOL) and social capital factors among “people living with HIV/AIDS” (PLWHA), in order to improve their quality of life and help them to release AIDS discrimination. *Methods*: A cross-sectional survey with 225 PLWHA was done in Nanchang, China, between January and June of 2015. Questionnaires consisted of a socio-demographic questionnaire, Personal Social Capital Scale and Medical Outcomes Study HIV Health Survey. To identify social capital factors influencing QOL among PLWHA, *t*-test and multiple linear regression were used as statistical tools. The analysis of data was conducted using SPSS 22.0 with a significant value of *p* < 0.05. *Results*: The scores of total social capital, bonding social capital and bridging social capital were 23.68 ± 5.55, 14.11 ± 3.40 and 9.46 ± 3.43 respectively. The scores of Physical Health Summary (PHS) and Mental Health Summary (MHS) were 51.88 ± 7.04 and 49.29 ± 6.60. Multiple linear regression analysis showed that age (B = −0.137, *p* = 0.020), income (B = 0.2170, *p* ≤ 0.001), employment (B = 0.112, *p* = 0.043) and bonding social capital (B = 0.178, *p* = 0.001) had significant effects on PHS. Bonding social capital was the most important influencing factor for MHS (B = 0.506, *p* < 0.001). There was no significant relationship between bridging social capital and QOL (*p* > 0.05). *Conclusions*: The PLWHA community has low social capital and a poor QOL in Nanchang. Bonding social capital had a positive impact on the QOL of PLWHA. There is an urgent need to build a better social support system based on bonding social capital for PLWHA. It is worth further exploring to identify how to make full use of bridging social capital for improving QOL among PLWHA.

## 1. Introduction

Acquired immunodeficiency syndrome (AIDS) has been regarded as primarily a serious global public health crisis, which has attracted worldwide attention [1]. Since the first report of HIV-infection incidence in China in 1985, the rate of HIV infection has increased to more than 30% per year and viral infection has gradually spread from high-risk groups to the general population [2]. By 30 September 2008, there were a total of 849,602 people living with HIV (PLWHA) or AIDS in China, including 497,231 PLWHA and 352,371 AIDS patients [3]. In addition, a total of 262,442 deaths were due to HIV infection and the ratio of male to female AIDS patients is 3.2:1 [3]. Nanchang is the capital city of Jiangxi Province located in the southern part of China, and it has been one of the most serious epidemic areas of HIV infection in the province [4,5]. According to the statistics from the Centers for Disease Control and Prevention (CDC) of Jiangxi Province, the total population of Jiangxi was 44.57 million and approximately 5.30 million people lived in Nanchang in 2016. There were 12,100 PLWHA and 3325 people died of HIV, which indicated HIV infection is a serious and also urgent health issue in the area [6]. Jiangxi Province has issued a number of state regulations on AIDS management, such as “Management Measures of AIDS Antiviral Treatment in Jiangxi Province”, “Three Free and One Subsidy” Policy for AIDS Treatment and so on, to control the prevalence of trend of HIV [7]. Furthermore, since 2015 the Finance Department of Jiangxi province has placed over 30 million Yuan to AIDS management program for free HIV testing, treatment, and follow-up health care [8].

Despite of the increased number of PLWHA, current treatment for HIV infection has improved and now more attention is being paid to the quality of life (QOL) and family care of PLWHA. Quality of life has been defined by WHO to be subjected to an individual’s perceptions of their positions in life where to live with particular relevance to the culture and value systems, and the life expectations, concerns, standards and life goals [9]. Due to socio-demographic, AIDS-related illness, social stigma and discrimination, and lack of family support, these reasons have generated negative emotions and caused lower QOL for PLWHA [10,11,12]. Although PLWHA are able to live much longer today, they may not always live in harmonious social atmosphere. How to improve QOL have become not only a medical problem, but also a huge social task. Therefore, it is important and necessary to look for more effective approaches to help PLWHA enhance their QOL. The concept of social capital may have a special impact on AIDS care and anti-AIDS discrimination for PLWHA.

The term social capital dates back to the 20th century, but it has not been conclusive until now. In much of the literature on health-related behavior, social capital is defined as a social network that is enduring, credible, mutually beneficial, and resource-rich for its members [13]. A large body of literature has proven that the less social an individual’s existing social capital, the more likely he/she is to behave in a health risk manner with unhealthy physical condition and mental illness [14,15,16]. Moreover, social capital is of great benefit to the prevention of AIDS in the community [17,18]. Social capital can be classified according to its types (bonding and bridging), components (structural and cognitive) and levels (micro and macro). In particular, social networks, social support, trust and participation are widely considered to be the core elements of social capital. This study employed the social capital scale established previously [13], which contains two parts: bridging social capital and bonding social capital. The Social Capital Scale focuses on the connection between individuals and society in personal social networks. Bonding social capital refers to a social network composed of members with common interests. Bridge social capital refers to a social network that connects different members through different social organizations or institutions [13].

Presently little study has been conducted assessing the effect of social capital on QOL among PLWHAs in China. Ma and her colleagues focus on the impact social capital and QOL at of individual-level, by measuring four dimensions of social capital: social network and ties, social support, social participation, and reciprocity and trust [19]. Lan’s study highlighted the cumulative impact of social capital on QOL, by measuring trust, social connection, and social participation [20]. While, our study aimed to assess the association between social capital and QOL by measuring bonding and bridging social capital. Bonding social capital can links individuals to each other basing on similar status and through their interests and mutual attraction, such as a family unit, friend relations, and neighborhood relations. Bridging social capital could also provide the linkage to people of different status through social groups and organizations [21,22]. Therefore, in this study, we particularly employed the Personal Social Capital Scale (PSCS) [23] as the main method to measure social capital for the PLWHA in China.

In this study, we designed to understand the relationship between social capital and QOL. Social capital includes bonding social capital and bridging social capital. QOL was divided into physical and mental health.

## 2. Materials and Methods

### 2.1. Study Population and Sample

This study was approved by the Medical Ethics Committee of the First Affiliated Hospital of Nanchang University on 4 March, 2014. To respect and protect the privacy of each participant, the surveys were conducted anonymously. Before the investigation, respondents expressed a verbal understanding of these issues and signed consent forms. All PLWHA who were registered in CDC database of the Donghu and Qingyunpu districts in Nanchang from January to June in 2015 were interviewed in this study. For Donghu district is located in the north of Nanchang and Qingyunpu district is in the south. Besides, they were the biggest CDC in Nanchang, which could treat PLWHA. Therefore they could represent the PLWHA’s situation of Nanchang well. The health status of PLWHAs was assessed by medical doctors in the CDC.

### 2.2. Inclusion and Exclusion Criteria

#### 2.2.1. Inclusion Criteria

(1)People who aged 16 years or older;(2)People who were diagnosed as PLWHA by doctors;(3)People who were treated in Donghu CDC or Qingyunpu CDC between January and June of 2015;(4)People who were alive and registered in Donghu CDC or Qingyunpu CDC.

#### 2.2.2. Exclusion Criteria

(1)People who aged ≤15 years old;(2)People who have mental diseases;(3)People who can’t express their own opinion well, such as those who can’t communicate with others;(4)People who have poor compliance.

### 2.3. Questionnaire

Questionnaires were consisted by socio-demographic questionnaire, personal social capital scale (PSCS) and Medical Outcomes Study HIV Health Survey (MOS-HIV) [24]. The questionnaire experts who have extensive and experience in survey study and knowledge in HIV and AIDS designed the study and oversaw and provided guidance the study and study progress.

The socio-demographic questionnaires included gender, age, education level, place of residence, marital status, average income of family and AIDS-related information, such as HIV-infected route, HIV-status disclosure, and CD4+ cell count. In China, people who are 16 years old or above are defined as adults and they can fill in the questionnaires without others effects. Besides, in this study, no person was student and the income was divided into two levels “stable and unstable”. If people who had got salary for more than 6 moths continuously, then, we defined him had a stable income. Moreover, “HIV-status disclosure” refers to respondents’ HIV-status disclosure to people except your family members.

PSCS includes two dimensions, bonding capital and bridging capital [13]. The Appendix A is the PSCS questionnaire. According the calculation scale, the instrument consisted of 10 items (Cap1 to Cap10) containing a total of 42 sub-items. A 5 point Likert scale with 1= ‘none’ or a few to 5 = ’all’ or a lot was employed assess all 42 sub-items. The test score for the first 5 items were calculated by computing the scores of 6 sub-items, and the subtotal was divided by 6. The score of items 6, 7, 8 and 9 were calculated by summarizing the 2 sub-item scores and the subtotal was divided by 2. The score of the last item was calculated by summarizing the 4 sub-item scores and then dividing the subtotal by 4. Upon obtaining item scores, the bonding social capital score was computed by adding the scores of item 1 to 5 together; bridging social capital score was calculated by adding the scores of items 6 to item 10 together. The total social capital score is the summary of the bonding and bridging social capital scores. All dimensions are scored on a scale from 0 to 25, and total score ranges from 0 to 50. Lower and higher scores indicated less and more social capital, respectively. The Personal social capital scale was shown to have good reliability and validity and the Cronbach’s α reliability for the scale was 0.77–0.87 [13].

Quality of life was assessed according to MOS-HIV with the simplified Chinese version covering 35 items [25], which focus on 10 health domains, including eight multi-item domains (general health, physical function, role function, cognitive function, pain, mental health, energy/fatigue, and health distress) and two single-item domains (social function and quality of life). An additional single-item inquiry was also included in the survey for health transition. Initial item scores for each domain was calculated and then changed into a 0–100 scale. Higher item scores signifying wellbeing and better functioning. Two factor-analysis-based summary scores: physical health summary (PHS) and mental health summary (MHS) score were also calculated according to the methods described previously [26]. Reliability and validity of the simplified Chinese version of the MOS-HIV questionnaire was acceptable to the Cronbach’s α of 0.69–0.87 [25].

### 2.4. Sample Size

In this study, a cluster sampling method was used in this survey, and we used Deff to adjust the method of calculating sample quantity. The formula as follows:$$n={\left(\frac{Z_{\alpha /2}\mathrm{CV}}{\varepsilon }\right)}^{2}$$

According to previous studies, the average score of the QOL of PLWHA is 46.81 [27]. In addition, the Standard Deviation is 10.58 and *α* = 0.05. The relative error “ε” required for this investigation was less than 5%, and the *n* we calculated is 80. The Deff used to be 1 to 3 in other surveys. In this survey, we set Deff as 2 and then 160 people were needed. Adding 10% to 15% of the missing or invalid questionnaires, the sample size should be 176 to 184 cases. The final sample size of this project is 225, which are far more than 184.

### 2.5. Data Collection and Data Analysis

The investigators were postgraduate students from the School of Public Health at Nanchang University and the staff from the CDC of the Donghu and Qingyunpu districts. Before filling in the questionnaires, we obtained the informed consent of the respondents. Besides, the respondents filled out the questionnaire anonymously to protect themselves from the risk of privacy exposure.

We used EpiData 3.0 software (The EpiData Association, Odense, Denmark) to entry data. Moreover, all descriptive and inferential statistics were conducted using SPSS 22.0 (SPSS Inc., Chicago, IL, USA). Proportions, means and standard deviations (SDs) were used for descriptive analysis of data. Both *t*-test and multiple linear regression analysis were used to identify the association between social capital and QOL among PLWHA. The significance level was set at *p* < 0.05.

### 2.6. Quality of Data

Professor Zhaokang Yuan of Nanchang University has extensive knowledge and field experience in survey studies and provided all the necessary training to the interviewers to ensure everyone understood the study, all the questions are clear, and how to get the survey study done effectively prior to survey starting. Furthermore, each interviewer needs to focus on one respondent at a time to ensure that no information was missed or unclear. The response rate was 97.8% with the support of CDC staffs and respondents.

## 3. Results

### 3.1. Socio-Demographics Features

This study included 184 males (81.8%) and 41 females (18.2%). Most of the subjects lived in the city (96.4%) and 44.9% respondents were married. There are about 65.8% respondents were educated at the senior high school level or above, 63.6% respondents were employed and 72.0% respondents had stable income. The most common transmission mode was through the sexual route of transmission (88.4%) and the majority were undisclosed HIV-status (80.9%) (Table 1).

### 3.2. Social Capital Scores

The scores of total social capital, bonding social capital, and bridging social capital were 23.56 ± 5.58, 14.11 ± 3.40, and 9.46 ± 3.43, respectively. The scores of total social capital and the two dimensions were significantly lower than general population (*p* < 0.05) (Table 2).

### 3.3. Effects of Demographic Characteristics on PHS and MHS

As shown in Table 3, PLWHAs who were unmarried, divorced or widowed, got a lower PHS (*t* = −14.248, *p* < 0.001). In addition of education, income, employment and CD4+ count were also the important variables that could significantly affect the PHS (*p* < 0.05). As for MHS, unmarried, divorced or widowed (*t* = −17.346, *p* < 0.001), lower income (*t* = −13.015, *p* < 0.001), unemployed (*t* = −11.796, *p* < 0.001) and the low CD4+ count (*p* < 0.05) would lead to lower MHS for the PLWHAs.

### 3.4. Quality of Life Score

The scores of 11 domains including general health, physical function, role function, social function, cognitive function, pain, mental health, energy/fatigue, health distress, quality of life, health transition were determined to be 59.27 ± 18.60, 84.41 ± 19.23, 70.58 ± 28.40, 76.27 ± 20.79, 60.33 ± 20.56, 84.24 ± 16.26, 75.98 ± 12.74, 77.56 ± 15.85, 76.67 ± 16.11, 63.11 ± 18.15, 63.11 ± 17.53, respectively. The scores for PHS and MHS were 51.88 ± 7.04 and 49.29 ± 6.60, respectively, among 225 PLWHAs.

### 3.5. Factors Associated with Quality of Life and Its Domains

To assess the association between QOL and the independent variables, multiple linear stepwise regression analysis was used. This study defined QOL score as the dependent variable. According to Table 3, variables such as demographics and CD4+ cell count were included in multiple linear regression analysis. The variable assignment for QOL and associated factors are shown in Table 4.

Variables such as age (B = −0.137, *p* = 0.020), bonding social capital (B = 0.178, *p* = 0.001), income (B = 0.217, *p* < 0.001) and employment (B = 0.112, *p* = 0.043) had a significant effect on physical health. Besides, compared to 0~199 CD4+ count, the 300–499 CD4+ count and above 400 CD4+ count had a positive effects on physical health scores (B = 0.185, 0.369, *p* < 0.001). The influencing factors in mental health were bonding social capital score (B = 0.506, *p* < 0.001), marital status (B = 0.304, *p* < 0.001) and income (B = 0.130, *p* = 0.005), indicating that bonding social capital was the most important influencing factors for mental health (Table 5).

## 4. Discussion

Quality of life is an essential measurement to consider in the overall health of PLWHA. Previous reports showed that the overall QOL were generally poor among PLWHAs [27,28]. The scores of PHS and MHS in this study were 51.88 ± 7.04 and 49.29 ± 6.60, respectively, which were lower than the normal people in China [29]. The QOL and its domains among the participants were better than AIDS patients in parts of China, except the domain of quality of life [30].

Consistent with a previous study [31], we found that lower CD4+ cell count was associated with lower PHS score. A higher CD4+ cell count reflected a healthier PLWHA status. The study also found that PLWHA who were employed with stable incomes had higher PHS and MHS scores since they had more suitable living environment. Also the PLWHA had a healthier body, which could be helpful to reduce mental stress caused by HIV infection. Physical health is a key element in maintaining a job since PLWHA will likely lose their jobs if their clinical symptoms appear [11]. In addition, married or cohabitation status will get courage from their spouse, which help them get out of some illnesses and have a higher MHS and PHS.

This study also showed that bonding social capital score has positive effects on MHS and PHS, which is consistent with Dong’s report indicating that patient’s family plays an important role in providing PLWHA with mental support and companionship [32]. This could be due to that love and company from family members help them to reduce any fear of ridicule, discrimination and death. In addition, more support and assistance from friends and companions are also essential for the PLWHA by providing confiding partners and emotional support, which greatly relieves the psychological pressure of PLWHA. Furthermore, being in the same situation, they may have similar feelings, and are much easier to understand each other. This also referred in Binagwaho’s study that in Rwanda, PLWHAs are always accompanied by their friends and relatives when they come to the hospital for treatment [27]. This means that company is very important for PLWHA’s daily lives no matter where it comes from. Another report further supported the importance of peer support in improving the quality of PLWHA life in clinical III and IV antiretroviral therapy [33].

The results showed that bridging social capital has no significant effect on physical and mental health among PLWHA in Nanchang, which might be attributed to: (a) to improve present HIV/AIDS prevention, treatment, and care services, Chinese government has started different programs such as “Four Frees and One Care” and “Task-Shifting” [34,35]. However, these approaches were far from meeting the current demand for the HIV/AIDS services and care; (b) non-governmental organizations (NGO) are very few in China and they cannot operate as independently as those in western countries. In addition, this study showed that PLWHA had little contact with organizations and communities by communicating with the participants. Most of these PLWHA are afraid of their HIV-status disclosure, even though they hoped to get more support and care from these organizations. Thus, it is important to explore the full use of bridging social capital for these patients in future.

Therefore, the study brought attention to local society in improving the QOL of PLWHA through bonding social capital, including building better social networks and strengthening social support based on family members, relatives, neighborhoods and friends, making full use of bridging social resources, especially communities. The model of social care could be organized and coordinated by local CDC, community health service staff, clinical staff, volunteers, family members, relatives, friends and colleagues to take care of PLWHA. More group activities are clearly important and necessary to improve the QOL of PLWHA locally and nationally.

Several limitations in this study should be addressed. First, the study only looked at one city and the ethnic diversity of the participants was not considered, so it cannot be generalized to other regions in China. Future studies would be carried out in various parts of China. Second, this study was conducted using retrospective research, and it may cause recall bias when we investigate PLWHA who remember some events unclearly. Finally, other confounding factors, such as viral load, sexual orientation, drinking and smoking were not considered in the questionnaires. For further exploration, we would take these factors into consideration.

## 5. Conclusions

This study reveals that PLWHA has low social capital and a poor QOL in Nanchang as compared to general population in China. The bonding social capital has a positive effect on PLWHA’s QOL including PHS and MHS. There is an urgent need to build a better social support system based on bonding social capital for PLWHA. Our findings suggest that the importance and essentiality of more depth study in order to figure out how to make full use of bridging social capital for improving QOL among PLWHAs.

## Figures and Tables

**Table 1 ijerph-16-00276-t001:** Demographic characteristics of 225 PLWHAs (people living with HIV/AIDS).

Variables		Total	Proportion (%)
Gender	Male	184	81.8
	Female	41	18.2
Place of residence	Urban	217	96.4
	Rural	8	3.6
Marital status	Married/cohabitation	101	44.9
	Others (Unmarried/divorced/widowed)	124	55.1
Education level	Primary school or below	34	15.1
	Junior high school	43	19.1
	Senior high school or above	148	65.8
Income	Unstable	63	28.0
	Stable	162	72.0
Employment	Employed	143	63.6
	Unemployed	82	36.4
HIV-infected route	Through sexual transmission	199	88.4
	Through blood transmission	10	4.4
	Others	16	7.2
HIV-status disclosure	Yes	43	19.1
	No	182	80.9
CD4+ count(cells/mm^3^)	<200	101	44.9
	200–299	28	12.4
	300–399	28	12.4
	≥400	68	30.3

**Table 2 ijerph-16-00276-t002:** Scores of social capital and its dimension among 225 PLWHA ((people living with HIV/AIDS)) and general population.

Item	Full Score	Respondents Score (X ± S)	General Population Score (X ± S)	*t*-Value	*p*-Value
Bonding social capital	25	14.11 ± 3.40	15.15 ± 3.01	−2.878	0.004
Bridging social capital	25	9.46 ± 3.43	10.76 ± 3.37	−3.445	0.001
Total social capital	50	23.56 ± 5.58	25.90 ± 5.25	−3.684	<0.001

**Table 3 ijerph-16-00276-t003:** Effects of demographic characteristics on PHS (Physical Health Summary) and MHS (Mental Health Summary).

Variables	PHS (X ± S)	*t*/F-Value	*p*-Value	MHS (X ± S)	*t*/F-Value	*p*-Value
Gender						
Male	52.04 ± 6.88			49.57 ± 6.60		
Female	51.17 ± 7.74	0.716	0.475	48.02 ± 6.53	1.365	0.174
Place of residence						
City	51.99 ± 7.09			49.40 ± 6.65		
Countryside	48.86 ± 4.91	−1.240	0.216	46.31 ± 4.64	−1.301	0.195
Marital status						
Married/cohabitation	57.17 ± 4.37			54.78 ± 3.94		
Others	47.57 ± 5.73	−14.248	<0.001	44.82 ± 4.67	−17.346	<0.001
Education level						
Primary school or below	48.64 ± 9.03 ^bc^			48.43 ± 5.80		
Junior high school	52.01 ± 6.60 ^a^			49.53 ± 6.67		
Senior high school or above	52.58 ± 6.46 ^a^	4.488	0.012	49.47 ± 6.78	0.344	0.709
Income						
Stable	55.11 ± 4.80			51.99 ± 5.10		
Unstable	43.57 ± 4.65	−16.338	<0.001	42.35 ± 4.70	−13.015	<0.001
Employment						
Employment	55.70 ± 4.76			52.37 ± 5.27		
Unemployment	45.22 ± 5.16	−15.397	<0.001	43.92 ± 5.11	−11.796	<0.001
HIV-infected route						
Through blood	53.38 ± 7.29			51.64 ± 6.75		
Through sexual	51.83 ± 6.88			49.16 ± 6.53		
Others	51.51 ± 9.03	0.251	0.779	49.42 ± 7.60	0.674	0.511
HIV-status disclosure						
Yes	52.09 ± 6.31			48.81 ± 6.71		
No	51.83 ± 7.21	−0.214	0.831	49.40 ± 6.59	0.530	0.597
CD4+ cell						
<200	45.89 ± 5.00 ^bcd^			44.04 ± 4.71 ^bcd^		
200–299	52.60 ± 3.46 ^acd^			51.87 ± 2.91 ^ad^		
300–399	56.29 ± 2.47 ^abd^			51.23 ± 3.54 ^ad^		
≥400	58.66 ± 3.45 ^abc^	141.909	<0.001	55.23 ± 4.74 ^abc^	93.918	<0.001

Multiple comparison tests with LSD. ^a^: Statistically significant compared to the first group. ^b^: Statistically significant compared to the second group. ^c^: Statistically significant compared to the third group. ^d^: Statistically significant compared to the forth group. *p* < 0.05.

**Table 4 ijerph-16-00276-t004:** Factors associated with QOL (Quality of Life) among PLWHAs (people living with HIV/AIDS) and their assignment.

Factors	Variables	Assignment
X_1_	Gender	Male = 0 (referent), Female = 1
X_2_	Age	Continuous variable
X_3_	Marital status	Others (Unmarried/divorced/widowed) = 0 (reference), Married/cohabitation = 1
X_4_	Education level	Senior high school or above = 0 (reference), Junior high school = 1, Primary school or below = 2
X_5_	Income	Stable = 0 (reference), Unstable = 1
X_6_	Employment	Employment=0 (reference), Unemployment = 1
X_7_	CD4+ cell count	<200 = 0 (referent), 200–299=1, 300–399 = 2, ≥400 = 3
X_8_	Bonding social capital score	Continuous variable
X_9_	Bridging social capital score	Continuous variable
X_10_	Total social capital score	Continuous variable
Y_1_	QOL	PHS score
Y_2_		MHS score

**Table 5 ijerph-16-00276-t005:** Factors associated with QOL based on multiple linear regression analysis.

Variables		Nonstandard Coefficient		Standard Coefficient (B)	*t*-Value	*p*-Value	95% CI
		b	S.E.				
PHS							
	Age	−0.064	0.027	−0.137	−2.340	0.020	(−0.117, −0.010)
	Bonding social capital score	0.369	0.108	0.178	3.425	0.001	(0.157, 0.581)
	Income	3.390	0.870	0.217	3.899	<0.001	(1.676, 5.104)
	Marital status	0.441	0.762	0.031	0.579	0.563	(−1.060, 1.943)
	Employment	1.634	0.804	0.112	2.034	0.043	(0.050, 3.218)
	CD4+ 200–299	0.796	0.978	0.037	0.814	0.416	(−1.131, 2.724)
	CD4+ 300–499	3.935	1.020	0.185	3.856	<0.001	(1.924, 5.946)
	CD4+ ≥400	5.649	0.989	0.369	5.713	<0.001	(3.700, 7.597)
	Primary school or below	−0.348	0.682	−0.018	−0.510	0.611	(−1.691, 0.996)
	Junior high school	0.778	0.604	0.044	1.289	0.199	(−0.411, 1.968)
MHS							
	Bonding social capital score	0.983	0.096	0.506	10.213	<0.001	(0.793, 1.173)
	Marital status	4.032	0.680	0.304	5.930	<0.001	(2.692, 5.372)
	Income	1.912	0.679	0.130	2.814	0.005	(0.573, 3.251)
	CD4+ 200–299	1.149	0.868	0.058	1.324	0.187	(−0.561, 2.860)
	CD4+ 300–499	−0.665	0.898	−0.033	−0.741	0.460	(−2.434, 1.105)
	CD4+ ≥400	1.090	0.872	0.076	1.250	0.213	(−0.629, 2.810)

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
