# Peer review of "Social Capital Associated with Quality of Life among People Living with HIV/AIDS in Nanchang, China"

_ijerph, 2019, doi:10.3390/ijerph16020276_

Reviewer 1 Report

The topic is one of importance given the high number of presentations to health services that are related to concerns on HIV/AIDS. If conducted with academic rigor, this article has the potential to be of value for practitioners and policymakers around the prevalence, cost and prevention of complications for people with this health problems. Furthermore, in my opinion the topic and premise of the study would sit well within the journal to which it was submitted. The authors should be commended for undertaking this study, however, there are a major concerns with the manuscript that require attention prior to publication. These will be discussed below relative to the sections of the manuscript. 

TITLE

The title is correct as it reflects correctly the objective of the work.

SUMMARY

Correct

INTRODUCTION: 

The introduction is poor, needs to present a better rationale for the study and the methodology employed. Also, the research question itself is sound, however the topic is not strongly introduced. No clear explanation is given to what constitutes health problems in Nanghnag, is it based on  all people in China?. Additional information and prevalence would benefit the reader (and several further sources are available prevalence is higher in  females or in males , but only one source. Overall, the introduction would benefit from a broader literature review and more detail on what the problem is, how much it impacts on HIV/AIDSand the economic implications of this would add depth to the intro. Further, include the work hypothesis in this section

MATERIALS AND METHODS: 

There is no information about validation of the questionnaire of quality of life included in the analysis including some explanation about the domains Also, include description of this questionnaire, its reliability and validity and the actual measurements. 

Thus, neither appear information related with inclusion and exclusion criteria.  The study design is a cross-sectional study of ramdom sampling method, where the study was conducted in the hospital or in the outpatient center?.

Please include the date and code register number of ethics committee

RESULTS: 

The results in basis of the used method are correct.

DISCUSSION: 

I am struggling to make sense of some of this, I am afraid it needs extensive revision. What are the clinical and non clinical implications of your study? How this will inform future larger studies?

CONCLUSION:

These conclusions need to be softened, modified a in order to reflect only the study findings.

Author Response

Response to Reviewer 1 Comments

The topic is one of importance given the high number of presentations to health services that are related to concerns on HIV/AIDS. If conducted with academic rigor, this article has the potential to be of value for practitioners and policymakers around the prevalence, cost and prevention of complications for people with this health problems. Furthermore, in my opinion the topic and premise of the study would sit well within the journal to which it was submitted. The authors should be commended for undertaking this study, however, there are a major concerns with the manuscript that require attention prior to publication. These will be discussed below relative to the sections of the manuscript.

Point 1: The title is correct as it reflects correctly the objective of the work.   

Response 1: Thanks.

Point 2: The introduction is poor, needs to present a better rationale for the study and the methodology employed. Also, the research question itself is sound, however the topic is not strongly introduced. No clear explanation is given to what constitutes health problems in Nanchang, is it based on all people in China?. Additional information and prevalence would benefit the reader (and several further sources are available prevalence is higher in females or in males, but only one source. Overall, the introduction would benefit from a broader literature review and more detail on what the problem is, how much it impacts on HIV/AIDS and the economic implications of this would add depth to the intro. Further, include the work hypothesis in this section

Response 2: We thank the reviewer for the excellent comment and have extensively revised the introduction accordingly (lines 44-95).

Point 3: There is no information about validation of the questionnaire of quality of life included in the analysis including some explanation about the domains Also, include description of this questionnaire, its reliability and validity and the actual measurements.

Response 3: We agreed with the reviewer and have added the information to the revised manuscript (lines 146-158).

Point 4: Thus, neither appear information related with inclusion and exclusion criteria. The study design is a cross-sectional study of random sampling method, where the study was conducted in the hospital or in the outpatient center?

Response 4: We thank the reviewer for pointing this deficiency out and this information has now added to the revised manuscript (lines 108-120).

Point 5: Please include the date and code register number of ethics committee

Response 5: This information has been included in revised manuscript now accordingly (Line 98-99)

Point 6: I am struggling to make sense of some of this, I am afraid it needs extensive revision. What are the clinical and non clinical implications of your study? How this will inform future larger studies?

Response 6: We thank the reviewer for these comments and have revised the manuscript extensively to address the concerns accordingly (Lines 237-255 and 267-273). 

Point 7: These conclusions need to be softened, modified a in order to reflect only the study findings.

Response 7: We thank the reviewer for the suggestion and have made the change accordingly (lines 282-287).

Author Response

Response to Reviewer 2 Comments

The manuscript deals with a potentially interesting topic. As is however, the manuscript is poorly written and needs major revisions for its potential contribution to emerge. Two major issues need to be addressed.

Point 1: The first relates to the lack of a coherent and concise theoretical framework from within which the study’s necessity (and the research hypotheses) will draw. Thus, I would suggest that the authors try to build a tight and informative theoretical framework, and one that incorporates the wealth of literature relating to the social capital effects on individuals’ status and perceptions of health.

Response 1: We thank the reviewer for this excellent comment and suggestion and have revised the introduction part extensively according to the suggestion (lines 44-95).

Point 2: The second major concern regards the empirical methods and the information offered by the study. This part needs to be considerably strengthened in order to provide a solid empirical basis for analysis and so the information acquired through it might be considered valid. Also a more clear presentation of methods should be followed.

The choice of the case study area needs to be further explained and supported. National rates of infection should be mentioned and compared with that of the study area.

Response 2: We agreed with the reviewer and have added more information to address the concern (Lines 103-106, 108-120).

As for the National rates of infection, we have now included this in the Introduction part of revised manuscript.

Point 3: The authors need to provide more info on the questionnaire expert and his role in the organization of the study.

Response 3: This information has now been added to the revised manuscript accordingly (lines 123-125).

Point 4: The social capital constructs and items need to be included and clearly presented in a table, the place I would see proper for the presentation of basic descriptive statistics as well.

Response 4: We have included the questionnaire as appendix part at the last part of this study.

Point 5: If you have different groups then basic descriptive statistics per group should be mentioned.

Response 5: No, we didn’t have different groups.

Point 6: The empirical results as also presented in a confusing manner. Please consider rephrasing.

Response 6: We thank the reviewer for this comment and would appreciate for more specific concern.

Point 7: The discussion part of the study should ideally focus on bringing together the extant literature in the field and the findings of the present study.

Response 7: We have revised the Discussion part extensively with more references about social capital effect on PLWHAs (lines 237-255, 267-273) accordingly.

Point 8: The manuscript needs an English spelling and grammar check.

Response 8: We have revised the manuscript to remove the typos.

Point 9: Some parts of the text should be re-organized (e.g. part 2.1.2. should be part of 2.1). The excess use of subtitles, e.g. for a paragraph, is not helpful.

Response 9: We agreed with the reviewer and have made the changes.

Round  2

Reviewer 1 Report

I do not have any further comments and/or suggestions to authors. Thank you very much for the effort and implementation of suggestions to the manuscript. 

Author Response

Response to Reviewer 1 Comments

Point 1: I do not have any further comments and/or suggestions to authors. Thank you very much for the effort and implementation of suggestions to the manuscript.

Response 1: Thank you so much.

Reviewer 2 Report

The authors have made great effort to improve their manuscript alongside incorporating the review comments. However there are still improvements that need to made with regard to presentation and in depth analysis of key concepts such as the social capital notion and their use in the current analysis. I would suggest that the authors avoid personal comments and use citations even for the work of authors and coauthors. in preparing a neutral text the authors should provide citations for all material used in the study or clarify personal communications if they used them to acquire relevant material. I did not see the descriptive statistics of the used variables.

Author Response

Response to Reviewer 2 Comments

Point 1: The authors have made great effort to improve their manuscript alongside incorporating the review comments. However there are still improvements that need to made with regard to presentation and in depth analysis of key concepts such as the social capital notion and their use in the current analysis. I would suggest that the authors avoid personal comments and use citations even for the work of authors and coauthors. in preparing a neutral text the authors should provide citations for all material used in the study or clarify personal communications if they used them to acquire relevant material. I did not see the descriptive statistics of the used variables.

Response 1: We thank the reviewer for the excellent comment and have extensively revised the introduction accordingly (lines 71-80; 200-204).

Round  3

Reviewer 2 Report

The authors have made an effort to appropriately revise their manuscript. However, earlier comments have not been incorporated in the revised manuscript, and particularly the english language editing (e.g. the content of the quesion in cap.1 is not clear at all). 

I would expect methods to be based on earlier literature. Besides, the authors did not revise certain sentences to appropriately present methods etc. in a neutral way (e.g Dr. Chen .....).

It is my view that the manuscript is difficult to follow due to syntax errors that should be addressed.

Author Response

Response to Reviewer 2’ Comments

Point 1: However, earlier comments have not been incorporated in the revised manuscript, and particularly the English language editing (e.g. the content of the question in cap.1 is not clear at all). 

Response 1: We thank the reviewer for the comment and have revised the introduction according to the suggestion (lines 76-78). We understand that social capital can be classified according to its types (bonding and bridging), components (Structural and Cognitive), and levels (Micro and Macro). Social networks, social support, trust, and participation are widely considered to be the core elements of social capital.

Point 2: I would expect methods to be based on earlier literature. Besides, the authors did not revise certain sentences to appropriately present methods etc. in a neutral way (e.g Dr. Chen .....). 

Response 2: We acknowledged this comment from the reviewer but have used Dr. Chen's social capital scale for this study for two main reasons Firstly, because PSCS has good reliability and validity, there is no other published social capital scale in China; and secondly, because previous research has focused on social capital, social support, trust, participation, and other social capital dimension, which is very suitable for this study addressing the current state of social capital from the perspective of the two dimensions of bonding and bridging.

Point 3: It is my view that the manuscript is difficult to follow due to syntax errors that should be addressed.

Response 3: We have tried our best to correct the syntax errors in the manuscript so as to make it more accessible accordingly.